# Novel Antimicrobial Peptides Designed Using a Recurrent Neural Network Reduce Mortality in Experimental Sepsis

**DOI:** 10.3390/antibiotics11030411

**Published:** 2022-03-18

**Authors:** Albert Bolatchiev, Vladimir Baturin, Evgeny Shchetinin, Elizaveta Bolatchieva

**Affiliations:** 1Department of Clinical Pharmacology, Stavropol State Medical University, 355000 Stavropol, Russia; prof.baturin@gmail.com; 2Department of Pathophysiology, Stavropol State Medical University, 355000 Stavropol, Russia; ev.cliph@rambler.ru; 3Department of Anatomy, Stavropol State Medical University, 355000 Stavropol, Russia; elizavetabolat@gmail.com

**Keywords:** neural network, LSTM, RNN, antibiotic resistance, antimicrobial peptide, peptide design, carbapenem-resistance, sepsis

## Abstract

The search and development of new antibiotics is relevant due to widespread antibiotic resistance. One of the promising strategies is the de novo design of novel antimicrobial peptides. The amino acid sequences of 198 novel peptides were obtained using a generative long short-term memory recurrent neural network (LSTM RNN). To assess their antimicrobial effect, we synthesized five out of 198 generated peptides. The PEP-38 and PEP-137 peptides were active in vitro against carbapenem-resistant isolates of *Klebsiella aerogenes* and *K. pneumoniae*. PEP-137 was also active against *Pseudomonas aeruginosa*. The remaining three peptides (PEP-36, PEP-136 and PEP-174) showed no antibacterial effect. Then the effect of PEP-38 and PEP-137 (a single intraperitoneal administration of a 100 μg dose 30 min after infection) on animal survival in an experimental murine model of *K. pneumoniae*-induced sepsis was investigated. As a control, two groups of mice were used: one received sterile saline, and the other received inactive in vitro PEP-36 (a single 100 μg dose). The PEP-36 peptide was shown to provide the highest survival rate (66.7%). PEP-137 showed a survival rate of 50%. PEP-38 was found to be ineffective. The data obtained can be used to develop new antibacterial peptide drugs to combat antibiotic resistance.

## 1. Introduction

In recent decades, humankind has faced the global problem of antimicrobial resistance (AMR), the main reason for which is natural evolutionary selection due to the excessive use of antimicrobials in medicine and agriculture [1,2,3]. Widespread use of antibiotics during the COVID-19 pandemic is likely to further accelerate the AMR spread [4]. According to a recent systematic review, even though no more than 7% of COVID-19 patients had bacterial co-infection, 72% of patients received antibiotics [5]. It is obvious that the most effective solution to the AMR problem is the development of new antimicrobial drugs. But in recent decades, the number of newly approved antibiotics was very limited [6]. Pharmaceutical companies are reluctant to invest in the development of new antibacterials because drug development is very expensive, and the potential profits may not cover the costs due to the rapid AMR development [7]. From this point of view, it is necessary to look for such compounds to which bacteria cannot form resistance or form it no faster than after 20–25 years, as long as the actual drug patent remains valid. Perhaps then the development of new antibiotics will become attractive for investments.

In recent years, of great interest are the so-called antimicrobial peptides (AMPs), which are produced by all living species. AMPs are known to have pronounced antibacterial, antiviral, antifungal, antiparasitic, and antitumor effects [8]. However, unfortunately, AMPs are much more expensive to synthesize than small molecules; moreover, some peptides are rapidly degraded when used in vivo, which can negatively affect their pharmacokinetic properties [9].

However, it is important to note that the market for peptide drugs is now actively growing [10]; moreover, the likelihood of developing resistance to AMPs is much lower than to conventional antibiotics [11]. The mechanism of action of AMPs is based on direct action on the bacterial cell wall via membrane permeabilization [12,13]. This makes AMPs equally effective against susceptible and multidrug-resistant bacteria [6]. Previously, we examined the strategy of using AMPs in combination with conventional antibiotics and showed that the antibiotic resistance phenotype of bacteria does not affect the effectiveness of AMPs [14]. In addition, it was also shown that natural human AMPs in combination with rifampicin and aminoglycosides demonstrate synergistic action against methicillin-resistant *staphylococci* and carbapenem-resistant strains of *E. coli*, respectively. Recently we showed that AMPs encapsulated in silicon nanoparticles can be an effective tool to develop topical AMP-based treatments [15], but this method doesn’t work for systemic administration. An alternative approach to the use of AMPs can be the design of completely new peptides with antimicrobial activity. Such a de novo design of AMPs could be a potentially effective strategy for combating antibiotic resistance; moreover, it can be commercially attractive. Earlier, Müller et al. suggested using generative long short-term memory (LSTM) recurrent neural network (RNN) for combinatorial de novo peptide design [16], but this approach has not been experimentally validated.

The aim of this work was to experimentally evaluate the antibacterial activity of the new peptides designed by LSTM RNN against multidrug-resistant strains in vitro and in vivo. 

## 2. Results

### 2.1. Development of Novel Peptides with Potential Antimicrobial Effect

For combinatorial de novo antimicrobial peptide design, we used generative long short-term memory (LSTM) recurrent neural network (RNN), which was suggested by Müller et al. in 2018 [16]. RNN models identify patterns in consecutive data and generate new data from the analyzed context. Amino acid sequences of peptides are good consecutive input data for such machine learning models. Thus, training of generative models on the sequences of antimicrobial peptides can make it possible to design new peptides with unique amino acid sequences [16]. We downloaded the full code from https://github.com/alexarnimueller/LSTM_peptides (accessed on 29 January 2022). 

To train the RNN model, we used the amino acid sequences of AMPs from the APD3 database (https://aps.unmc.edu/, accessed on 29 January 2022) [17]. Unlike Müller et al., we used a different dataset—all-AMPs from the APD3 database, not only helical peptides. Peptides shorter than seven amino acid residues were excluded from the training set. The final training set comprised 3100 sequences with a mean sequence length of 33.6 ± 22.2 and a median length of 29 amino acid residues (file “Training dataset.csv” in the Appendix A).

We have chosen 200 as the number of sequences to sample from the model after training, but two of the sequences were shorter than seven amino acid residues. As such, finally, 198 novel peptide sequences were generated (Figure 1, file “Sampled peptides.csv” in the Appendix A). The generated 198 sequences were also compared to random sequences with the same amino acid distribution (as the training set) (file “random_sequences.fasta” in the Appendix A) as well as to manually generated, presumably helical sequences (file “helical_sequences.fasta” in the Appendix A) [16] (Figure 1). 

The generated 198 sequences were assessed using the CAMP AMP prediction tool (http://camp.bicnirrh.res.in/predict/, accessed on 29 January 2022) [18]. (the following verification algorithms were used: SVM, Random Forest and Discriminant Analysis). When using CAMP AMP we chose the sequences that had an AMP probability value of more than 0.950, predicted by all three algorisms. After that, we chose those peptides whose antimicrobial activity was predicted by all three algorithms simultaneously. After this rapid screening, we were left with 35 sequences with potential antimicrobial activity.

In the next stage of computer screening, we ran 35 sequences through the prediction algorithms proposed by B. Vishnepolsky and M. Pirtskhalava (https://dbaasp.org/ accessed on 29 January 2022) [19]: “Prediction of activity against specific microbial species”. The latter calculates the predictive value of activity or inactivity against various microorganisms: *Escherichia coli* ATCC 25922, *Pseudomonas aeruginosa* ATCC 27853, *Klebsiella pneumonia*, *Staphylococcus aureus* ATCC 25923. Thus, according to the prediction results, only the following peptides were active against all these strains: PEP-36, PEP-38, PEP-136, PEP-137, and PEP-187. It would be most rational to synthesize these five peptides whose activity was predicted for all microorganisms. However, subjectively, we decided not to synthesize PEP-187 and chose a different sequence—PEP-174; it was also predicted to be active against most bacteria except for *P. aeruginosa*. Finally, for the subsequent synthesis, we selected five out of 198 sequences (Table 1). All in silico screening results with amino acid sequences are available in the Appendix A “Peptides screening algorithms.xlsx”. 

### 2.2. PEP-38 and PEP-137 Are Active against Carbapenem-Resistant Gram-Negative Bacteria In Vitro

We investigated the antibacterial activity of five synthesized peptides (Table 1), which were selected from 198 amino acid sequences generated using LSTM RNN.

In our preliminary experiments, the synthesized peptides did not show activity against gram-positive bacteria; therefore, in this work for the in vitro experiments we used carbapenem-resistant isolates of *Klebsiella aerogenes* (*n* = 12), *K. pneumoniae* (*n* = 18), and *Pseudomonas aeruginosa* (*n* = 17). According to EUCAST breakpoints, all isolates of *Klebsiella spp*. were sensitive only to tigecycline, whereas the *P. aeruginosa* strains were sensitive only to ceftazidime/avibactam. Colistin sensitivity was not tested (this antibiotic is not included in routine microbiological testing in Russia). The range of concentrations of the studied peptides was from 0 to 32 μg/mL (the upper value was chosen empirically).

The PEP-36, PEP-136, PEP-174 peptides were not active against the studied isolates. The PEP-38 and PEP-137 peptides had a high level of antibacterial activity against the studied multidrug-resistant bacteria (Appendix A).

Minimum inhibitory concentration (MIC) of PEP-38 against *K. aerogenes* was 6 (4–8) μg/mL (hereinafter MIC values are presented as median, in brackets—first and third quartile); against K. pneumoniae, it was 8 (4–8) μg/mL. PEP-38 was not active against *P. aeruginosa*.

PEP-137 demonstrated a higher antimicrobial activity against the studied isolates. MIC against *K. aerogenes* was 2 (1–2.5) μg/mL; against *K. pneumoniae—*2 (1–4) μg/mL; against *P. aeruginosa—*2 (2–4) μg/mL. 

### 2.3. PEP-36 and PEP-137 Peptides Reduce Mortality in Experimental Sepsis

In an experimental murine model of *K. pneumoniae*-induced sepsis, we studied the effect of a single injection of PEP-38 and PEP-137 on the survival rate in comparison with the control (sterile saline solution); PEP-36 peptide was chosen as an additional control, since PEP-36 did not demonstrate the antimicrobial effect in vitro.

In the control group, the probability of survival was 0% on the third day after infection (Figure 2, Appendix A).

The most active peptide in the in vivo experiments, PEP-137, had significant differences from the control group (*p* = 0.02694): the survival proportion by the end of the observation period was 50% (Figure 2, Appendix A).

PEP-38, which was also effective in vitro, did not reduce mortality in an experimental model of sepsis in vivo and did not significantly differ from the control group (Figure 2, Appendix A).

Unexpected results were obtained on survival rate in the analysis of the effect of the PEP-36 peptide, which was completely inactive in vitro. After a single injection of this peptide, eight out of 12 mice survived by the end of the observation period—a probability of survival was 66.7% (*p* = 0.00051 compared to the control) (Figure 2, Appendix A). 

### 2.4. PEP-36, PEP-38 and PEP-137 Peptides Have Similarity in Their Spatial Structure 

For the rapid and simple modeling of the spatial structure of the synthesized peptides, the recently developed AlphaFold v2.1.0 algorithm was applied. Since we did not need to analyze the physicochemical properties of the obtained spatial structures in-depth, as a simple example for comparison, we used the structure of human cathelicidin LL-37 from the Protein Data Bank (PDB ID: 2K6O) [20]. Figure 3 shows obvious similarities in the spatial structures of all studied molecules and LL-37. 

### 2.5. PEP-36 and PEP-38 Are Potentially Less Toxic to Red Blood Cells

When we ran the amino acid sequences of all five peptides through the HAPPENN classifier [21], which demonstrates an accuracy of more than 85%, it was shown that PEP-137 and PEP-134 have the highest hemolytic potential with PROB scores 0.962 and 0.832, respectively. Not effective PEP-136 demonstrated the lowest toxicity (PROB = 0.008). Peptides PEP-36 and PEP-38 had a low red blood cell lysing potential with PROB values 0.236 and 0.191, respectively. 

### 2.6. Molecular Dynamics Simulation of PEP-36, PEP-38 and PEP-137 Interaction with Bacterial Membrane

To study the peptide-membrane interaction, each peptide was placed at the “top” of *E. coli* membrane and molecular dynamics (MD) simulation was performed for 225 ns (Figure 4). Figure 4 depicts that at the end of 225 ns, PEP-38 was able to penetrate the lipid bilayer, whereas PEP-36 and PEP-137 were not penetrated. The helical structure of PEP-36 was unraveled to render the globular form. The penetration and unravelling of helical structure are more visible in Figure 5. The root mean squared plot of peptides in Appendix A shows the unravelling of PEP-36 helical structure in membrane surface. We can assume that the disruption of the helical structure of PEP-36 makes it difficult to penetrate the bacterial membrane. 

According to MD results, among three peptides, PEP-38 is more promising in terms of penetrating the bacterial membrane, and hence it could be capable of producing membrane lysis. As PEP-137 has a greater number of charged amino acids, it could not penetrate the membrane, since the charged amino acids favor the hydrophilic environment in membrane surface. 

## 3. Discussion

In this work, we evaluated the method suggested by Müller et al. [16], for the first time experimentally (in vitro and in vivo), to generate novel peptides with presumable antimicrobial activity. We slightly modified the input data for the neural network to use all 3100 antimicrobial peptides (with a length of more than seven amino acid residues) from the APD3 database (https://aps.unmc.edu/, accessed on 29 January 2022) [17]. The neural network generated 198 sequences that were run through various in silico screening systems, after which, five peptides were chosen for synthesis (PEP-36, PEP-38, PEP-136, PEP-137 and PEP-174).

The first stage of the experimental screening was the study of the minimum inhibitory concentrations of these peptides by the method of serial dilutions. PEP-38 was found to be active against *K. aerogenes* and *K. pneumoniae*. PEP-137 was more effective against *Klebsiella spp*., as well as against *P. aeruginosa*. The PEP-36, PEP-136 and PEP-174 peptides were not active against these bacteria. It should be noted. However, a limitation of this work was that we did not study the effectiveness of concentrations above 32 μg/mL.

The data obtained prompted us to test the effectiveness of the PEP-38 and PEP-137 peptides in the simplest experimental model of sepsis in mice. Moreover, as an additional control, we randomly selected PEP-36, which was not active in vitro. It should be noted that we used a simple and easily reproducible sepsis model with a single injection of the studied peptides at a dose of 100 μg/mouse.

Surprisingly, the PEP-36 peptide was found to be the most effective—the animal survival rate was 66.7%. PEP-137 showed a survival rate of 50%. PEP-38 was proven to be ineffective. 

In their work on LL-37, Guangshun Wang et al., using nuclear magnetic resonance structural analysis, identified a short three-turn amphipathic helix rich in positively charged side chains, which helps to effectively compete for anionic phosphatidylglycerols in bacterial membranes [20].

The modeled spatial structures of the novel peptides are very similar to the LL-37 molecule. It seems likely that the helix-rich structure of the PEP-36, PEP-38, and PEP-137 peptides may be an important contributor to the demonstrated antimicrobial effect. It remains unclear why PEP-38 was active in vitro but did not affect mortality in experimental sepsis; it may be necessary to investigate larger doses of this peptide and other routes of administration.

It is also unclear why PEP-36 did not have antimicrobial activity in vitro but was the most effective in vivo. This can be explained by modifications after entering the body (i.e., proteolytic cleavage of some amino acids), or this peptide might have an immunomodulatory effect. To clarify the nature of the data obtained, more research is required. Earlier, in the *P. aeruginosa*-induced sepsis model, epinecidin-1 (Epi-1; by *Epinephelus coioides*) due to its antimicrobial and pronounced immunomodulatory effect has been shown to reduce mortality in mice [22,23,24]. Epi-1 enhances the production of IgG antibodies by activating the Th2-cell response [23], reduces the level of tumor necrosis factor-alpha by reducing the level of endotoxins [22]. 

It is very intriguing that molecular dynamics simulations have shown that the PEP-38 must be the most promising candidate. It turned out that the opposite was shown in vivo—PEP-38 was not effective, while PEP-36 and PEP-137 reduced mortality. 

The data obtained in this work require a wide range of additional experiments: the study of pharmacodynamics and pharmacokinetics, toxicity, as well as the study of the combined action of the obtained peptides with conventional antibiotics.

Thus, we conducted a simple and easily reproducible study with an experimental assessment of the feasibility of using the generative long-term memory recurrent neural network to generate novel peptides that demonstrate antimicrobial activity in vitro and reduce mortality in experimental sepsis in vivo. The peptides obtained can be used to develop new antibacterial drugs for the treatment of infections caused by carbapenem-resistant, gram-negative bacteria. 

## 4. Materials and Methods

### 4.1. Peptides 

Peptides were synthesized on a commercial basis by AtaGenix Laboratories (Wuhan, China) using Fmoc solid-phase synthesis and reverse phase HPLC purification. Peptide identity was confirmed by electrospray mass spectrometry. Purity (>95%) was determined by HPLC. Amino acid sequences and some physicochemical properties of the synthesized peptides are presented in Table 1. 

### 4.2. Bacteria 

The strains of bacteria were isolated in 2021 from patients in the intensive care unit of the Stavropol State Regional Clinical Hospital. Identification and determination of antibiotic resistance of bacterial isolates were carried out using the disk diffusion method as part of a routine microbiological study in accordance with the European Committee on Antimicrobial Susceptibility Testing (EUCAST) protocols in the Department of Clinical Microbiology of the Center of Clinical Pharmacology and Pharmacotherapy [25]. 

### 4.3. In Vitro Study of Antibacterial Activity 

The study of the antimicrobial activity of the synthesized peptides was performed by the standard broth dilution method [26] according to the EUCAST guidelines [27]. Briefly, pure bacterial cultures were cultured on a solid nutrient media (mannitol salt agar, BioMedia, Russia). From a fresh morning culture, we prepared a suspension in sterile saline corresponding to a McFarland turbidity standard of 0.5 (which is equivalent to 1–2 × 10^8^ CFU/mL). The resulting suspension was dissolved in the BBL^TM^ Mueller-Hinton broth (Becton, Dickinson and Company, Franklin Lakes, NJ, USA) to obtain an inoculum with an approximate concentration of 5 × 10^5^ CFU/mL. The inoculum (100 μL per well) was added to the wells of a sterile 96-well microtiter plate with a U-shaped bottom (Medpolymer, St. Petersburg, Russia). After that, serial two-fold dilutions of tested peptides (100 μL per well) were added to the wells. There were also sterility control wells (Mueller-Hinton broth only, without bacteria) and growth control wells (bacterial inoculum without peptides). Then, the plates were incubated in a thermostat at 37 °C. After 18–20 h, the minimum inhibitory concentration (MIC) values were determined. The MIC was taken as the minimum peptide concentration at which there was no visual growth in the corresponding well [26]. The peptides were dissolved in the Mueller–Hinton broth. The experiments with each peptide and bacterial isolate were performed in triplicate (in different plates). MIC values were calculated as median values of three independent replicates (calculations are presented in Appendix A). 

### 4.4. Murine Experimental Model of Sepsis 

To assess the effectiveness of the studied peptides, a murine model of lethal generalized infection was used. The model of sepsis makes it possible to perform a simple and rapid screening assessment of the effectiveness of new antimicrobial agents in vivo [22,28]. 

Animal studies were approved by the local ethics committee of the Stavropol State Medical University (Protocol No. 95 dated 18 February 2021) and were performed in accordance with the Code of Ethics of the World Medical Association (Declaration of Helsinki, EU Directive 2010/63/EU for animal experiments). Animal studies are reported in compliance with the ARRIVE guidelines [29]. In order to generate animal groups of equal size, randomization of animals between groups was carried out. 

The ICR (CD-1) laboratory mice (females with an average weight of 30 g) were maintained in the vivarium of the Stavropol State Medical University. The mice (four animals per cage) were housed in temperature-controlled rooms at 24 °C and 50–60% humidity with a 12 h/12 h light/dark cycle and water and food availability ad libitum. 

The animals were injected with bacterial suspensions of one of the carbapenem-resistant isolates of *K. pneumoniae* (isolate #7, Appendix A) prepared from a fresh morning culture in accordance with the McFarland turbidity standard of 15 (which corresponds to an approximate concentration of 4.5 × 10^9^ CFU/mL). To determine the turbidity of the suspension, a DEN-1 densitometer (Biosan, Latvia) was used. The bacterial suspension and peptides were dissolved in sterile saline and injected intraperitoneally (the total volume of injected fluid did not exceed 250 μL per mouse).

In the experiment, there were four groups of 12 mice in each: the first group was a control one (it received saline); the second group was an additional control (it received PEP-36, which was not active in vitro); the third group received PEP-38; the fourth group received PEP-137. At t = 0 min, all animals were infected with *K. pneumoniae*: 150 μL of a 4.5 × 10^9^ CFU/mL suspension (approx. 6.75 × 10^8^ CFU per mice). 30 min after infection, the mice received a single injection of peptides at a dose of 100 μg. Mortality was assessed every 24 h for 5 days. Moribund animals were killed humanely to avoid unnecessary distress. 

### 4.5. Modeling the Structure of Novel Peptides

To model highly accurate structures of synthesized peptides, we used the recently published algorithm AlphaFold [30] (a slightly simplified version of AlphaFold v2.1.0 available online, DeepMind Technologies Limited, London, UK). To generate the images, the PyMOL Molecular Graphics System Version 2.5.2 (Schrödinger, New York, NY, USA) was used. The .pdb files of PEP-36, PEP-38, and PEP-137 peptides are available in Appendix A). 

### 4.6. In Silico Predicting of Hemolytic Potential of Novel Peptides 

To compute the potential toxicity of the designed peptides against red blood cells, we used online, bioinformatic web server, HAPPEN (https://research.timmons.eu/happenn, accessed on 29 January 2022). This classifier has been recently shown to achieve best-in-class performance, with cross-validated accuracy of 85.7% and Matthews correlation coefficient of 0.71 [21]. HAPPENN calculates the PROB score, that is, the normalized sigmoid score and ranges between 0 and 1. 0 is predicted to be most likely non-hemolytic, 1 is predicted to be most likely hemolytic.

### 4.7. Statistical Analysis 

Calculations of MICs (median, first, and third quartile) were performed in Microsoft Excel (Appendix A). Survival analysis by the Kaplan–Meier method and log-rank (Mantel–Cox) test was performed using GraphPad Prism version 9.2.0 for macOS (GraphPad Software, San Diego, CA, USA, www.graphpad.com, accessed on 29 January 2022). 

### 4.8. Molecular Dynamics Modeling 

To investigate the interaction of top three peptides (PEP-36, PEP-38, PEP-137) with bacterial membrane, we performed the molecular dynamics (MD) simulation of the peptides with Gram-negative inner membrane. Most of the antimicrobial peptides are able to penetrate the outer membrane of Gram-negative bacteria, but their interaction with inner-membrane is important for bactericidal activity [31]. Hence, Gram-negative inner-membrane and peptide systems were built for the study purpose. Membrane-peptide systems for PEP-36, PEP-38, and PEP-137 were generated with the membrane builder plugin of CHARMM-GUI [32]. The lipid bilayer contained POPE, POPG, and TMCL1 (cardiolipin) lipids in the proportion of 75:20:5 to represent the membrane of Gram-negative bacteria, *E. coli* [33]. Each system was neutralized by 0.15 mM KCl and solvated with TIP3P water model. 

All-atom MD simulations were performed using the GPU version of NAMD 2.14 [34] with the CHARMM36m [35] forcefield in an explicit water solvent. The minimization and equilibration of the membrane system were performed following the 6-step protocol as given by CHARMM-GUI [32]. Periodic boundary conditions were employed in all simulations. The long-range electrostatic interactions were treated using the particle mesh Ewald (PME) method [36], and the hydrogen atoms were constrained with the SHAKE algorithm [37]. A Nosé–Hoover Langevin piston method was used with a piston period of 50 fs and a decay of 25 fs to control the pressure. All simulations were run at 300 K temperature with Langevin temperature coupling and a friction coefficient of 1 ps^−1^. Each system was run for 225 ns of the production run. For calculations of hydrogen bonds (Appendix A), a cut-off distance of 3.5 Å and a cut-off angle of 30° were used. Visual molecular dynamics [38] were used to visualize and analyze the trajectories.

## 5. Conclusions

In this study, we evaluated the effectiveness of novel peptides generated using a recurrent neural network in vitro experiments and in a murine model of sepsis for the first time. The obtained data opens new possibilities for the design of new peptide-based antibiotics to combat carbapenem-resistant gram-negative bacteria. 

## Figures and Tables

**Figure 1 antibiotics-11-00411-f001:**
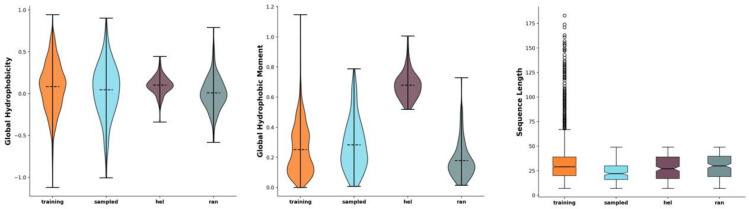
Comparison of peptide characteristics between the training data (Training, orange), the generated sequences (Sampled, blue), the pseudo-random sequences with the same amino acid distribution as in the training set (Ran, purple), and the manually created hypothetical amphipathic helices (Hel, green). The horizontal dashed lines represent the mean (violin plots) and median (box plots) values; the whiskers extend to the outermost non-outlier data points. Graphs from left to right: Eisenberg hydrophobicity, Eisenberg hydrophobic moment, and sequence length. The figure was generated using the modlAMP’s GlobalAnalysis.plot_summary method in Python [16].

**Figure 2 antibiotics-11-00411-f002:**
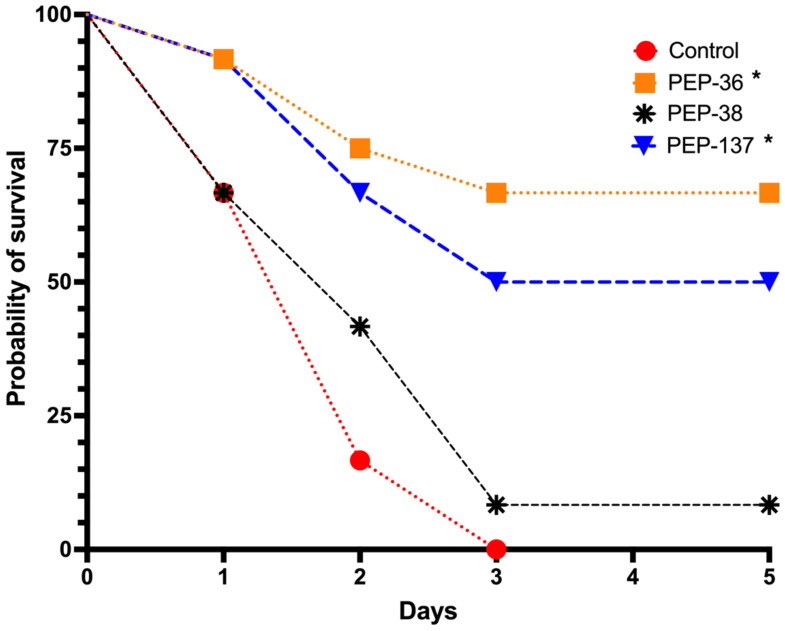
Comparison of probability of survival (%) in each group: control (sterile saline), PEP-36, PEP-38 and PEP-137. The peptides were injected once in a dose of 100 μg 30 min after infection of mice with 6.75 × 10^8^ CFU suspension of a carbapenem-resistant isolate of *K. pneumoniae*. *—significant differences from the control group using the Kaplan-Meier method and Log-rank (Mantel-Cox) test.

**Figure 3 antibiotics-11-00411-f003:**
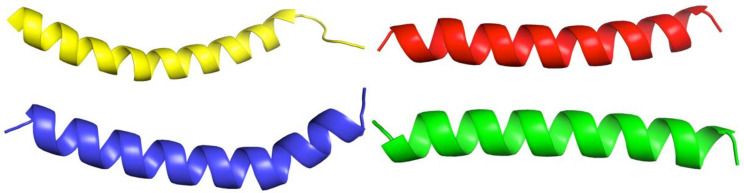
Visual comparison of the structures of LL-37 (yellow; PDB ID: 2K6O) and synthesized peptides: PEP-137—blue, PEP-38—red, PEP-36—green (by AlphaFold).

**Figure 4 antibiotics-11-00411-f004:**
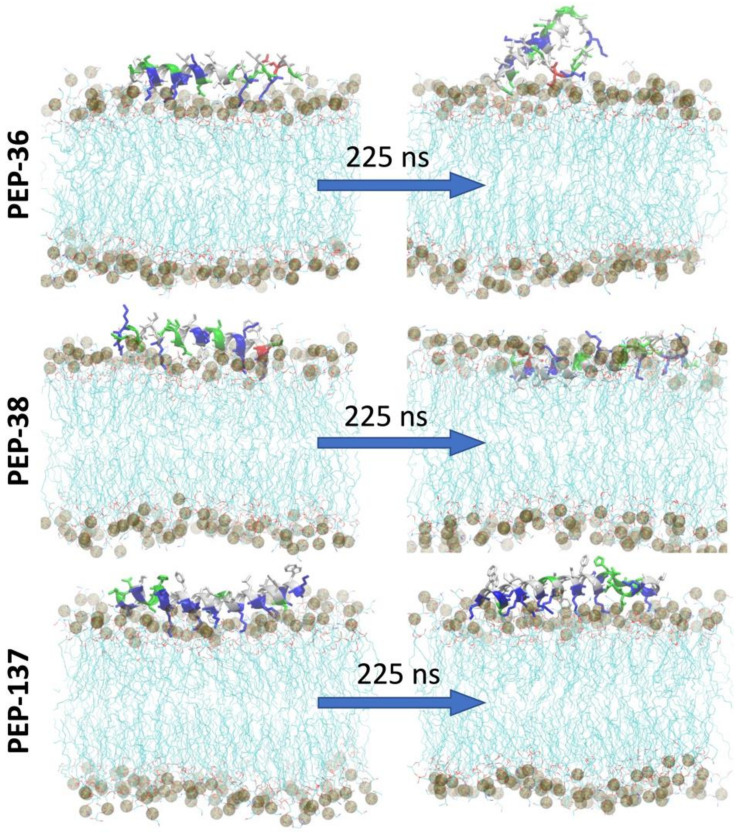
Initial and final snapshot of peptide-membrane system before and after 225 ns of MD simulation. The membrane is shown in lines (cyan) and the peptides are shown in helical structure (colored according to the type of amino acids). Phosphate atoms of the membrane are shown in gray color.

**Figure 5 antibiotics-11-00411-f005:**
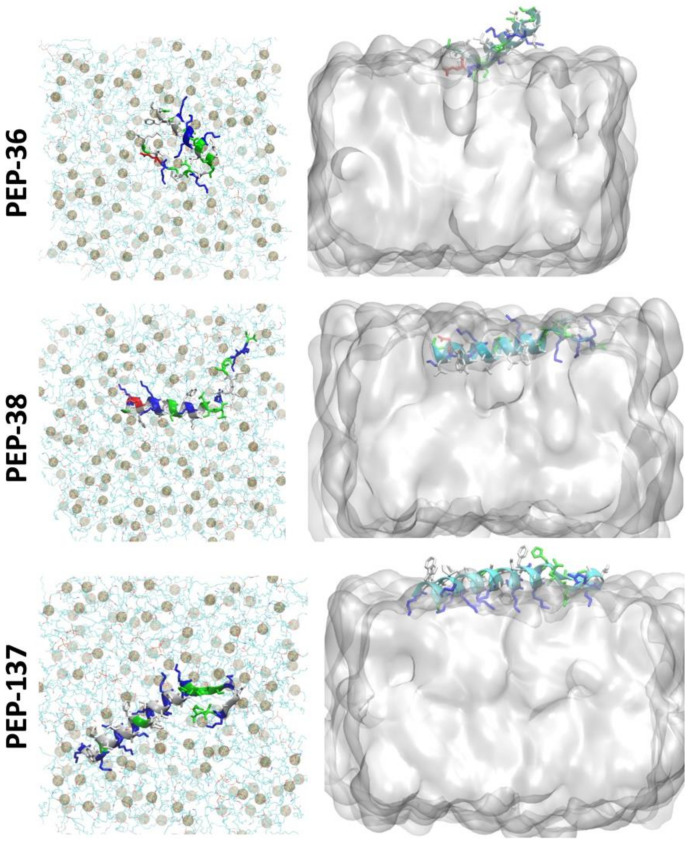
Top view of the peptide-membrane system (**left**) at 225 ns of MD simulation showing the unravelling of the helical structure. Phosphate atoms of the membrane are shown in gray color. Side view (**right**) showing the penetration of peptide at 225 ns.

**Table 1 antibiotics-11-00411-t001:** Amino acid sequences and characteristics of AMPs used.

Peptide	Amino Acid Sequence	Length	Molecular Weight	Charge	Hydrophobic Residues
PEP-36	GIFSKLAGKKIKNLLISGLKNIGKEVGM	28	2958	+5.0	43
PEP-38	GLKDWVKKALGSLWKLANSQKAIISGKKS	29	3156	+6.0	41
PEP-136	KWKLFKKIWSSVKLKS	16	2007	+6.0	44
PEP-137	KWKSFIKKLAKFGFKVIKKFAKKHGSKIAKNQ	32	3764	+12.1	41
PEP-174	GILSSFKGVLKGAGKNLLGSLKDKLKN	27	2786	+5.0	37

## Data Availability

Not applicable.

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
