# Peer review of "Novel Antimicrobial Peptides Designed Using a Recurrent Neural Network Reduce Mortality in Experimental Sepsis"

_antibiotics, 2022, doi:10.3390/antibiotics11030411_

Round 1

Reviewer 1 Report

In the article entitled " Novel antimicrobial peptides designed using recurrent neural network reduce mortality in experimental sepsis " authors performed the de novo peptide design and synthesis to determine novel peptides with anti-microbial properties against multi-drug resistance microorganisms. Though robust and unique the study have some serious flaw and the data and the inferences from the study is still inconclusive. The article is not yet ready for publication unless additional experiments are performed (and which must be performed) to answer some outstanding questions. Below are some of my major and minor comments:

Major Comments:

My major concern with this study is that most of the experiments and results are rudimentary and inconclusive. Though the authors tried to use the pipeline of LTSM-RNN to identify and design new peptide with antimicrobial properties, most of them showed variable Anti-microbial activity in vitro and in vivo. PEP36 was found to be inactive in vivo while PEP38 was in in vitro. Authors themselves admitted in the discussion section that the reason for this discrepancy is still unknown and will need for study to ascertain the same. Without this additional data the current work will be premature to publish.

Also, the authors performed some preliminary structural analysis. Again, this analysis is very rudimentary and didn't provide any valuable information to the study. Perhaps, more in depth structural analysis might provide the answer to AMP describe above.

Conducting additional experiments will help the authors not only to understand the bio and physiochemical properties of these designed peptides in their AMPs but will also aid them in designing novel Peptide with more potent anti-microbial properties.

Page 2, Line 58-60: Apart from the use of silicon based nanoparticles to administer AMPs topically what are the difference between design of AMPs between these two studies. In fact the following proposing statement of peptides with unique AMP suggest that this study didn't use AMPs with noval sequence. Also how the authors plan to deliver the AMPs they designed for systemic administration.

Minor Comments:

1) Abstract: In one or two sentences explain what the study is about in the beginning itself.

2) Page 1, Line 39: "do not.....after 20-30 years". Clarify this statement.

3) Page 2, Line 48: "Withal".. However should be the more appropriate word here. Also consider merging the line 48-49

4) Page 2, Line 55: " in this work" is redundant and can be removed. and replaced "shown" with "also shown"

5) Figure 2, Always provide the protein/peptide 3-dimensional structure in white background.

6) Alex T Müller et al: Only referring as Muller et al is sufficient. The first and middle name can be removed from main text.

Author Response

Thanks a lot for the review. We agree completely with your comments about our manuscript. 

  • Major comments

Response: we are not sure that the study is preliminary, but of course we have little experimental data. We did not set the task of a deep study of new peptides (at this stage). The main goal was to show that artificial intelligence (neural network) can generate unique sequences of antimicrobial peptides. Then it was necessary to conduct a very rapid screening of the effectiveness of these peptides in vitro and in vivo. That is, we needed to experimentally check whether the RNN-LSTM method works in general. And we showed for the first time that this technology can be used to obtain libraries of new peptides. Unfortunately, we no longer have funding and cannot conduct additional "wet" experiments. We can say with confidence that we are now awaiting the results of the examination of a new grant, after which we will conduct extended studies of the obtained peptides. 

However, we did small molecular dynamics studies modeling the interaction of all three peptides with the bacterial membrane and obtained some additional results, which we included in the new version of the manuscript. 

  • Page 2, Line 58-60: Apart from the use of silicon based nanoparticles to administer AMPs topically what are the difference between design of AMPs between these two studies. In fact the following proposing statement of peptides with unique AMP suggest that this study didn't use AMPs with noval sequence. Also how the authors plan to deliver the AMPs they designed for systemic administration.

The study we cite here we used natural antimicrobial peptides that were encapsulated in silicon nanoparticles.

As for the present manuscript, of course, we are still thinking about how we will solve the problem of delivering newly designed peptides. We have a few strategies that we can't reveal yet as we have a grant application for this topic.

Minor comments. 

  • 1) Abstract: In one or two sentences explain what the study is about in the beginning itself.

Added in the beginning of the abstract. 

  • 2) Page 1, Line 39: "do not.....after 20-30 years". Clarify this statement.

We specified it. We meant that in order for the development of antibiotics to become investment attractive, pharmaceutical companies need compounds to which resistance is slowly forming - so that the company can recoup the costs of development and research

  • 3) Page 2, Line 48: "Withal".. However should be the more appropriate word here. Also consider merging the line 48-49

Done. 

  • 4) Page 2, Line 55: " in this work" is redundant and can be removed. and replaced "shown" with "also shown"

Done. 

  • 5) Figure 2, Always provide the protein/peptide 3-dimensional structure in white background.

Done. 

6) Alex T Müller et al: Only referring as Muller et al is sufficient. The first and middle name can be removed from main text.

Done. 

Reviewer 2 Report

The manuscript entitled „Novel antimicrobial peptides designed using recurrent neural network reduce mortality in experimental sepsis” describes the identification and verification of antimicrobial peptides through combinatorial de novo peptide design. Based on a selection of AMP´s from the APD3 database and results obtained from their analysis, 5 peptides were selected for experimental verification in vitro and in vivo.

The study is interesting and could help to identify novel antimicrobial peptides. This is of relevance for identification of novel antibiotics.

However, there are several comments relating to this manuscript:

  • I would propose that the authors revise the conceptualization of their manuscript. Rather than starting the Results section with the five selected peptides, I would recommend to restructure the paper as follows: Section 4.1 of Materials & Methods should be presented in the Results section (until line 211). This helps the reader to understand how these five peptides were identified and selected. Then, the basis for the selection of the peptides and the description of the experiments should follow.
  • Section 2.1 describing the results against various bacterial isolates (lines 76-84) should be described in more detail, what bacteria were used and at which concentrations were the peptides tested, etc. This paragraph is too short and confusing.
  • Abbreviation MIC (line 79) should be written in full first in the Results section.
  • Line 93 should read: “The most active peptide in the in vivo experiments, PEP-137, ...”
  • The in vivo efficacy of the inactive peptide PEP-36 is interesting and possible explanations are discussed. However, it would be interesting to see whether a higher concentration (above 8 µg/ml) could also induce an antibacterial effect in vitro.

Author Response

Thank you very much for your review and important comments. 

  • I would propose that the authors revise the conceptualization of their manuscript. Rather than starting the Results section with the five selected peptides, I would recommend to restructure the paper as follows: Section 4.1 of Materials & Methods should be presented in the Results section (until line 211). This helps the reader to understand how these five peptides were identified and selected. Then, the basis for the selection of the peptides and the description of the experiments should follow.

Done.

  • Section 2.1 describing the results against various bacterial isolates (lines 76-84) should be described in more detail, what bacteria were used and at which concentrations were the peptides tested, etc. This paragraph is too short and confusing.

We have expanded this section for clarification. 

  • Abbreviation MIC (line 79) should be written in full first in the Results section.

Done. 

  • Line 93 should read: “The most active peptide in the in vivo experiments, PEP-137, ...”

Done. 

  • The in vivo efficacy of the inactive peptide PEP-36 is interesting and possible explanations are discussed. However, it would be interesting to see whether a higher concentration (above 8 µg/ml) could also induce an antibacterial effect in vitro.

We did not expect such a result. This is a completely random find. We will definitely investigate higher concentrations in more detail. We are just waiting for a new grant to finance these studies. 

Regarding the concentration of 8 µg/ml - we made a typo. The maximum concentration we tested was 32 µg/mL. We have corrected this defect. In supplementary materials, we presented these data in the form of tables S1-S3 when first applied the manuscript. 

Reviewer 3 Report

The manuscript of Bolatchiev et al. subjected to my revision describes the effect of five peptides to some drug-resistant bacteria in vitro and in vivo. In general, I like the biological part of the paper. The experiments are well designed, the results are clearly presented. Also, the english is well with only small typos found. However, some points rise my attention.

First of all, I do not understand, how actually the peptides were chosen. The neural networks are trained on some usually labelled examples (training set), the effectiveness of the training is validated (on validation set) with appropriate cost function, and then are used "on production" with unknown data (the test set). Here, the authors claim to train the network on over 3000 peptides. How was the training validated? What was the cost function? How were the data represented and normalized? What was the architecture of the network? What were the two layers used? What was the input, what was the output? In which framework was the network written? These is actually a selection of questions the authors should answer to describe their neural network. In particular, I don't understand, how the network was able to create a peptide sequence and why exactly 198. And also, there are 20 amino acids and peptides with median of 29 residues, which means at least 580 variables. These cannot fit into a neural network with 2 layers, and 256 neurons, as each neuron has only one tunable parameter?

Furthermore, I do not understand, how the final 5 peptides were chosen. I get the idea, that 198 peptides obtained somehow from the RNN were subjected to further analysis. But what were the criteria of the analysis? Did the authors chose the peptides active according to at least one criterium, or all? What is the criterium of being active?

Also, I do not understand the statistical analysis of the data, in particular the MIC calculation. The authors calculate the values on 12 isolates 3 times. But instead of calculating the median of the 12 isolates, they calculate the median of those 3 measurements. These leads to cases, when median of 16, 16, and 8 is 16, although this may be very far from the real value. Side note, the authors claim that the concentrations were up to 8 μg/ml, so why 16? And how was the final median calculated? As a mean of all medians? Please clarify. Also, I do not feel like the authors have enough statistics to calculate the quantiles.

I also do not understand, why the authors have chosen LL-37 as a comparison. And actually what is the analysis of the structure used for?

Finally, there are some small editorial errors:

  • line 232 "The study the antimicrobial" - missing "of", 
  • line 57 "staphylococci" should be in italics.

Also, I would recommend to add the values described in the section 2.1 to the table, instead of the molecular weight, or charge values, which are not discussed. By the way, what is charge +12.25?

Author Response

Thanks a lot for this recommendations and comments – they are very helpful to improve our work. 

  • First of all, I do not understand, how actually the peptides were chosen. The neural networks are trained on some usually labelled examples (training set), the effectiveness of the training is validated (on validation set) with appropriate cost function, and then are used "on production" with unknown data (the test set). Here, the authors claim to train the network on over 3000 peptides. How was the training validated? What was the cost function? How were the data represented and normalized? What was the architecture of the network? What were the two layers used? What was the input, what was the output? In which framework was the network written? These is actually a selection of questions the authors should answer to describe their neural network. In particular, I don't understand, how the network was able to create a peptide sequence and why exactly 198. And also, there are 20 amino acids and peptides with median of 29 residues, which means at least 580 variables. These cannot fit into a neural network with 2 layers, and 256 neurons, as each neuron has only one tunable parameter?

This is a very important note. And, unfortunately, in our team of authors, all experimental pharmacologists are not specialists in neural networks. We just took the "ready to use" neural network proposed by Muller and colleagues. The entire code of this neural network is available in the public domain at https://github.com/alexarnimueller/LSTM_peptides 

A. T. Müller, J. A. Hiss, G. Schneider, "Recurrent Neural Network Model for Constructive Peptide Design" J. Chem. Inf . Model. 2018, DOI: 10.1021/acs.jcim.7b00414. 

In the study, Muller et al. give a very detailed description of the entire architecture of this network. They presented a detailed description of their method with all the mathematical parameters.

In our work, we changed absolutely nothing in the neural network itself. We just expanded the training dataset. Muller et al. used 1554 peptides encompassing 7−48 amino acid residues with an average sequence length of 20.8 ± 7.7 (mean ± SD) and median = 21 residues. 

That is, we did not begin to give the method in detail, since it is very detailed and reliably presented in the work of Muller et al. This work was published by the Journal of Chemical Information and Modeling. 

Once again, we want to emphasize that we did not make any changes to the neural network itself. Our main task was to experimentally check whether the peptides obtained by artificial intelligence work in general or not. 

  • Furthermore, I do not understand, how the final 5 peptides were chosen. I get the idea, that 198 peptides obtained somehow from the RNN were subjected to further analysis. But what were the criteria of the analysis? Did the authors chose the peptides active according to at least one criterium, or all? What is the criterium of being active?

We have added the description of the selection criteria for the 5 peptides in more detail in Section 2.1. Also we added supplementary file “Peptides screening algorithms.xlsx” where we deeply described all of the in silico procedures. It is totally and easily reproducible. 

  • Also, I do not understand the statistical analysis of the data, in particular the MIC calculation. The authors calculate the values on 12 isolates 3 times. But instead of calculating the median of the 12 isolates, they calculate the median of those 3 measurements. These leads to cases, when median of 16, 16, and 8 is 16, although this may be very far from the real value. Side note, the authors claim that the concentrations were up to 8 μg/ml, so why 16? And how was the final median calculated? As a mean of all medians? Please clarify. Also, I do not feel like the authors have enough statistics to calculate the quantiles.

When writing the manuscript, we made a typo - we investigated concentrations up to 32 µg/ml (not 8 µg/ml). In supplementary information, we indicated this when the manuscript was first submitted (in tables S1-S3). In the description of the S3-table, we made the same typo. 

Indeed, for each isolate, we performed three independent measurements and calculated the median of these three measurements. The arithmetic mean, in our opinion, would be less reliable. We then calculated the median for each bacterial species. Similarly, we calculated values in previous work on natural antimicrobial peptides: https://www.mdpi.com/search?authors=bolatchiev&journal=antibiotics#:~:text=https%3A//doi.org/10.3390/antibiotics11010076   We also calculated first (25) and third (75) quartiles using simple Excel calculations. We have done this so that readers have a small idea of the "spread" of MIC values.   

  • I also do not understand, why the authors have chosen LL-37 as a comparison. And actually what is the analysis of the structure used for? 

Peptide LL-37 has an alpha-helical structure, which has been confirmed by solution nuclear magnetic resonance. We wanted to show that the new peptides are structurally similar to LL-37 (which has proven antimicrobial and other activities). That is, we wanted to visually demonstrate that, due to the similarity of their spatial structures, all these peptides can theoretically have similar effects.  

  • line 232 "The study the antimicrobial" - missing "of", 

Done. 

  • line 57 "staphylococci" should be in italics.

Done. 

  • Also, I would recommend to add the values described in the section 2.1 to the table, instead of the molecular weight, or charge values, which are not discussed. By the way, what is charge +12.25?

We calculated the charges using the algorithm https://pepcalc.com/ 

We believe that physico-chemical characteristics play an important role in the effectiveness of peptides, although we do not discuss them, we have presented this information in order to expand the general information - since we have created new peptides that have not been previously. +12.25 is a typo – corrected to 12.1. 

Round 2

Reviewer 1 Report

In the revised article authors have tried to adequately answer my previous concerns and also provided some initial in silico simulations analysis explaining the difference in mode of action of  the three peptides. The Manuscript is ready for publication after the some minors errors are taken care of:

1) Line 101: "algorisms" spell check

2)  MD simulation studies: Consider replacing the word "insert" with "penetrate". 

Author Response

Thanks a lot for review. 

1) Line 101: "algorisms" spell check

Done. 

2)  MD simulation studies: Consider replacing the word "insert" with "penetrate".

 Done. 

Reviewer 3 Report

The Authors indeed improved their description of protein selection after the use of RNN. However, the description of the RNN is far from acceptable. Therefore, I will state my concerns point by point:

  • What was the architecture of the RNN? Which layers were used?
  • In which framework the RNN was written?
  • What was the shape of the input data? Was it just a sequence? Was it normalized, or padded somehow?
  • Neural Networks operate on float vectors and matrices. How the sequences were encoded? Was it one-hot encoding vectors?
  • The authors claim, that the RNN was trained on 3100 sequences. How was the training efficiency measured? Do the authors have any validation set? In particular, the effect of the training is the trained Neural Network. The effect of the training is not the sequence. The pipeline is as follows: First, one trains the neural network, and once it is trained, one can use it to predict some features.
  • So the next question is, which set was used to train the network, and which was used to predict the sequences?
  • I guess the authors did not train the network, rather they used an already trained model on this set of 3100 sequences. How then such a RNN work? The sequences from the output are not part of the "training set", which means, they were somehow created by RNN. So why there are exactly 198 sequences? What was the criterium to select those? Was there any cutoff probability on how such sequence is going to be functional?
  • Again, the model the authors provide seems to be too small, as it can have less weights than the number of possible sequences. So it should not learn enough. So again, how the neural network was validated?

Actually, I could continue, but I think I made my point, that the description of the RNN usage is vague and does not allow to recreate such an experiment. Moreover, it contains errors, which cannot be present in the published version, like the "Upon completion of training, 198 novel peptide sequences were generated" - again, this sentence makes no sense. The effect of the training is the trained neural network, not the sequence. It is the trained RNN in turn can be used to predict the sequences, if some data are given to it.

In a conclusion, this manuscript cannot be published without extensive clarification of this part.

Possibly, the authors can write point by point, what did they do, and how did they use the RNN, and the details would be understandable from the context.

Author Response

  • Possibly, the authors can write point by point, what did they do, and how did they use the RNN, and the details would be understandable from the context.

We have tried to answer all your questions and points - from line 79 to line 142 you will find a more detailed description of our work, which we presented according to your recommendation. We have also added Figure 1 and Figure S6 for a more detailed presentation.

Moreover, we found raw files and logs after the work of the neural network and also presented them. We are sure that this will help readers to understand in more detail exactly how we set up the neural network. These files are also available now as supplementary files. 

We are very grateful to you for such a detailed review, as your recommendations helped us present our work in more detail for better understanding. 

Round 3

Reviewer 3 Report

My main objection for the last version was the description of the RNN model used. The authors made a substantial progress in this matter, however, I am still not satisfied. Before I will explain my point of view in details, let me notice, that in my opinion the authors make themselves a serious harm by forcing to make the use of RNN an important part of the manuscript. It would be much better if the authors stated from the beginning, that they used the methodology of Muller et al, with no change except for the training set.

However, the authors have chosen to include in their narration also building the RNN model, including selection of hyperparameters (like number of layers, which is actually three, not two). In fact, the model, and even the number of epochs is proposed in the Muller paper. Moreover, in the current version, the manuscript contains the Figure from the original paper with most of the caption copied, which raises my concern about the copyright. It does not say, that the Figure (and the caption) was copied with permition.

Concerning my specific remarks:

  • Apart from caption to Fig. 1, quite a large parts of methods is copied from Muler et al;
  • The number of layers is 3, as evidenced if not in the code, then when running the training (the Dense layer is also a layer).
  • The input data was the tensor of the dimension 49x22, there were 3100 such tensors;
  • You do not "solve" the loss function. You minimize it.
  • You do not calculate "average validation loss". You train untill loss is minimal. So calculating the mean value does not tell you anything. Nor the SD.
  • There is no S6 Figure in SI. But judging from Fig. S4 one may understand, what the authors meant by "average validation loss". If I am correct, the authors mean the "categorical cross-entropy", which in fact is weighted average. But then it should be just called "categorical cross-entropy".
  • Also, looking at S4, it seems that the authors have an overfitting to the training data starting from around 110 epochs. Maybe they should use some early stopping?
  • Even if the authors had minimal loss after 167 epochs (similar to Muller paper BTW), it is not a sufficient idicator to use those hyperparameters. When establishing the hyperparameters the authors should rather make a grid search of optimal ones.
  • The name of the crucial layer is spelled sometimes LSTM, sometimes LTSM (in abstract and introduction).

Also, I still do not know what is the "Charge 12.1". What is the fractional charge? And even if so, why the other charges are with different accuracy (e.g. 6 instead of 6.0)?

To sum up, the paper still contains quite a substantial number of flaws, with my largest concern now being the amount of the text copied from the referenced article. In my opinion, the safest option would be to stop underlying the RNN input, but only mention, that the authors used the method already published, trained on different dataset.

Author Response

Thank you so much, that was my best experience with manuscript revisions. I believe this revisions have made our paper much better. 

So, according to your final recommendation –

"...In my opinion, the safest option would be to stop underlying the RNN input, but only mention, that the authors used the method already published, trained on different dataset.." 

we have significantly reduced the description of the work of the neural network, but simply indicated that we used the previously published method by Muller et al. with our own dataset.

-- Also, I still do not know what is the "Charge 12.1". What is the fractional charge? And even if so, why the other charges are with different accuracy (e.g. 6 instead of 6.0)?

-- Response: we have corrected the calculated charge values in Table 1. 

Best wishes, 

Albert.